# TARGET PROPAGATION VIA REGULARIZED INVERSION

## ABSTRACT

Target Propagation (TP) algorithms compute targets instead of gradients along neural networks and propagate them backward in a way that is similar yet different than gradient back-propagation (BP). The idea was first presented as a perturbative alternative to back-propagation that may improve gradient evaluation accuracy when training multi-layer neural networks (Le Cun et al., 1989). However, TP may have remained more of a template algorithm with many variations than a well-identified algorithm. Revisiting insights of Le Cun et al. (1989) and more recently of Lee et al. (2015), we present a simple version of target propagation based on a regularized inversion of network layers, easily implementable in a differentiable programming framework. We compare its computational complexity to the one of BP and delineate the regimes in which TP can be attractive compared to BP. We show how our TP can be used to train recurrent neural networks with long sequences on various sequence modeling problems. The experimental results underscore the importance of regularization in TP in practice.

## 1 INTRODUCTION

Target propagation algorithms can be seen as perturbative learning alternatives to the gradient back-propagation algorithm, where virtual targets are propagated backward instead of gradients (Le Cun, 1986; Le Cun et al., 1989; Rohwer, 1990; Mirowski & LeCun, 2009; Bengio, 2014; Goodfellow et al., 2016). A high-level summary is presented in Fig. 1: while gradient back-propagation considers storing intermediate gradients in a forward pass, target propagation algorithms proceed by computing and storing approximate inverses. The approximate inverses are then passed on backward along the graph of computations to finally yield a weight update for stochastic learning.

Target propagation aims to take advantage of the availability of approximate inverses to compute better descent directions for the objective at hand. Bengio et al. (2013); Bengio (2020) argued that the approach could be relevant for problems involving multiple compositions such as the training of Recurrent Neural Networks (RNNs), which generally suffer from the phenomenon of exploding or vanishing gradients (Hochreiter, 1998; Bengio et al., 1994; **?**). Recently, empirical results indeed showed the potential advantages of target propagation over classical gradient back-propagation for training RNNs on several tasks (Manchev & Spratling, 2020). However, these recent investigations remain built on multiple approximations, which hinder the analysis of the core idea of TP, i.e., using layer inverses.

On the theoretical side, difference target propagation, a modern variant of target propagation, was related to an approximate Gauss-Newton method, suggesting interesting venues to explain the benefits of target propagation (Bengio, 2020; Meulemans et al., 2020). Previous works have considered approximating inverses by adding multiple reverse layers (Manchev & Spratling, 2020; Meulemans et al., 2020; Bengio, 2020). However, it is unclear whether such reverse layers actually learn layer inverses during the training process. Even if they were, the additional cost of computational complexity of learning approximate inverses should be carefully accounted for.

In this work, we propose a simple target propagation approach, revisiting the original insights of Le Cun et al. (1989) on the critical importance of the good conditioning of layer inverses. We define regularized inverses through a variational formulation and we obtain approximate inverses via these regularized inverses. In this spirit, we can also interpret the difference target propagation formula (Lee et al., 2015) as a finite difference approximation of a linearized regularized inverse. We propose a smoother formula that can directly be integrated into a differentiable programming framework.

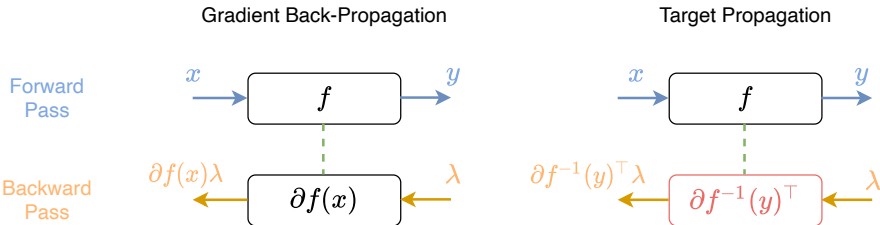

Fig. 1: Our implementation of target propagation uses linearization of gradient inverses instead of gradients in a backward pass akin to gradient back-propagation.

We detail the computational complexity of the proposed target propagation and compare it to the one of gradient back-propagation, showing that the additional cost of computing inverses can be effectively amortized for very long sequences. Following the benchmark of Manchev & Spratling (2020), we observe that the proposed target propagation can perform better than classical gradient-based methods on several tasks involving RNNs.

**Related work.** Many variations of back-propagation algorithms have been explored; see Werbos (1994); Goodfellow et al. (2016) for an extensive bibliography. Closer to target propagation, penalized formulations of the training problem have been considered to decouple the optimization of the weights in a distributed way or using an ADMM approach (Carreira-Perpinan & Wang, 2014; Taylor et al., 2016; Gotmare et al., 2018). Rather than modifying the backward operations in the layers, one can also modify the weight updates for deep forward networks by using a regularized inverse (Frerix et al., 2018). Wiseman et al. (2017) recast target propagation as an ADMM-like algorithm for language modeling and reported disappointing experimental results. Recently, in a careful experimental benchmark evaluation, Manchev & Spratling (2020) explored further target propagation to train RNNs, mapping a sequence to a single final output, in an attempt to understand the benefits of target propagation to capture long-range dependencies, and obtained promising experimental results. Another line of research has considered synthetic gradients that approximate gradients using an additional layer instead of using back-propagated gradients (Jaderberg et al., 2017; Czarnecki et al., 2017) to speed up the training of deep neural networks. Recently, Ahmad et al. (2020); Dalm et al. (2021) considered using analytical inverses to implement target propagation and blend it with what they called a gradient-adjusted incremental formula. Yet, an additional orthogonality penalty is critical for their approach to work. Recently, Meulemans et al. (2020) considered using as many reverse layers as forwarding operations. We focus here on the optimization gains of using target propagation that cannot be obtained by adding a prohibitive number of reverse layers. Finally, we do not discuss the biological plausibility of TP since we are unable to comment on this. We refer the interested reader to, e.g., (Bengio, 2020).

**Notations.** For $f : \mathbb{R}^p \times \mathbb{R}^q \to \mathbb{R}^d$, we denote $\partial_x f(x, y) = \left( \partial f^j(x, y) / \partial x_i \right)_{i,j} \in \mathbb{R}^{d \times p}$.

## 2 TARGET PROPAGATION WITH LINEARIZED REGULARIZED INVERSES

While target propagation was initially developed for multi-layer neural networks, we focus on its implementation for recurrent neural networks, as we shall follow the benchmark of Manchev & Spratling (2020) in the experiments. Recurrent Neural Networks (RNNs) are also a canonical family of neural networks in which interesting phenomena arise in back-propagation algorithms.

**Problem setting.** A simple RNN parameterized by $\theta = (W_{hh}, W_{xh}, b_h, W_{hy}, b_y)$ maps a sequence of inputs $x_{1:\tau} = (x_1, \ldots, x_\tau)$ to an output $\hat{y} = g_\theta(x_{1:\tau})$ by computing hidden states $h_t \in \mathbb{R}^p$ corresponding to the inputs $x_t$.

Formally, the output $\hat{y}$ and the hidden states $h_t$ are computed as an output operation following transition operations defined as

$$\hat{y} = c_\theta(h_\tau) := s(W_{hy} h_\tau + b_y),$$
$$h_t = f_{\theta,t}(h_{t-1}) := a(W_{xh} x_t + W_{hh} h_{t-1} + b_h) \quad \text{for } t \in \{1, \ldots, \tau\},$$

where $s$ is, e.g., the soft-max function for classification tasks, $a$ is a non-linear operation such as the hyperbolic tangent function, and the initial hidden state is generally fixed as $h_0 = 0$. Given samples of sequence-output pairs $(x_{1:\tau}, y)$, the RNN is trained to minimize the error $\ell(y, g_\theta(x_{1:\tau}))$ of predicting $\hat{y} = g_\theta(x_{1:\tau})$ instead of $y$.

As one considers longer sequences, RNNs face the challenge of exploding/vanishing gradients $\partial g_\theta(x_{1:\tau})/\partial h_t$ (Bengio & Frasconi, 1995); see Appendix A for more discussion. We acknowledge that specific parameterization-based strategies have been proposed to address this issue of exploding/vanishing gradients, such as orthonormal parameterizations of the weights (Arjovsky et al., 2016; Helfrich et al., 2018; Lezcano-Casado & Martınez-Rubio, 2019). The focus here is to simplify and understand target propagation as a backpropagation-type algorithm using RNNs as a workbench. Indeed, training RNNs is an optimization problem involving multiple compositions for which approximate inverses can easily be available. The framework could also be potentially applied to, e.g., time-series or control models (Roulet et al., 2019).

Given the parameters $W_{hh}, W_{xh}, b_h$ of the transition operations, we can get approximate inverses of $f_{\theta,t}(h_{t-1})$ for all $t \in \{1, \dots, \tau\}$, that yield optimization surrogates that can be better performing than the ones corresponding to regular gradients. We present below a *simple version* of target propagation based on *regularized inverses* and *inverse linearizations*.

**Back-propagating targets.** The idea of target propagation is to compute virtual targets $v_t$ for each layer $t = \tau, \dots, 1$ such that if the layers were able to match their corresponding target at time $t$, i.e., $f_{\theta,t}(h_{t-1}) \approx v_t$, the objective would decrease. The final target $v_\tau$ is computed as a gradient step on the loss w.r.t. $h_\tau$. The targets are then back-propagated using an approximate inverse[1] $f_{\theta,t}^{-1}$ of $f_{\theta,t}$ at each time step.

Formally, consider an RNN that computed $\tau$ states $h_1, \dots, h_\tau$ from a sequence $x_1, \dots, x_\tau$ with associated output $y$. For a given stepsize $\gamma_h > 0$, we propose to back-propagate targets by computing

$$v_\tau = h_\tau - \gamma_h \partial_h \ell(y, c_\theta(h_\tau)), \tag{1}$$

$$v_{t-1} = h_{t-1} + \partial_h f_{\theta,t}^{-1}(h_t)^\top (v_t - h_t), \quad \text{for } t \in \{\tau, \dots, 1\}. \tag{2}$$

The update rule (2) blends two ideas: i) regularized inversion; ii) linear approximation. We shall describe below that our update (2) allows us to interpret the "magic formula" of difference target propagation in Eq. 15 of Lee et al. (2015) as 0th-order finite difference approximation, while ours is a 1st-order linear approximation. We shall also show that (2) puts in practice an insight from Bengio (2020) suggesting to use the inverse of the gradients in the spirit of a Gauss-Newton method.

Once all targets are computed, the parameters of the transition operations are updated such that the outputs of $f_{\theta,t}$ at each time step move closer to the given target. Formally, the update consists of a gradient step with stepsize $\gamma_\theta$ on the squared error between the targets and the current outputs, i.e., for $\theta_h \in \{W_{hh}, W_{xh}, b_h\}$,

$$\theta_h^{\text{next}} = \theta_h - \gamma_\theta \sum_{t=1}^{\tau} \partial_{\theta_h} \|f_{\theta,t}(h_{t-1}) - v_t\|_2^2 / 2. \tag{3}$$

As for the parameters $\theta_y = (W_{hy}, b_y)$ of the output operation, they are updated by a simple gradient step on the loss with a stepsize $\gamma_\theta$.

## 2.1 REGULARIZED INVERSION

To explore further the original idea of Le Cun et al. (1989), we consider using the variational definition of the inverse,

$$f_{\theta,t}^{-1}(v_t) = \underset{v_{t-1} \in \mathbb{R}^p}{\operatorname{argmin}} \|f_{\theta,t}(v_{t-1}) - v_t\|_2^2 = \underset{v_{t-1} \in \mathbb{R}^p}{\operatorname{argmin}} \|a(W_{xh}x_t + W_{hh}v_{t-1} + b_h) - v_t\|_2^2. \tag{4}$$

As long as $v_t$ belongs to the image $f_{\theta,t}(\mathbb{R}^p)$ of $f_{\theta,t}$, this definition recovers exactly the inverse of $v_t$ by $f_{\theta,t}$. More generally, if $v_t \notin f_{\theta,t}(\mathbb{R}^p)$, Eq. (4) computes the *best approximation* of the

---

[1]In the following, to ease the presentation, we abuse notations and denote approximate inverses by $f_{\theta,t}^{-1}$.

inverse in the sense of the Euclidean projection. When one considers an activation function $a$ and $\theta_h = (W_{hh}, W_{xh}, b_h)$, the solution of (4) can easily be computed.

Formally, for the sigmoid, the hyperbolic tangent or the ReLU, their inverse can be obtained analytically for any $v_t \in a(\mathbb{R}^p)$. So for $v_t \in a(\mathbb{R}^p)$ and $W_{hh}$ full rank, we get

$$f_{\theta,t}^{-1}(v_t) = (W_{hh}^\top W_{hh})^{-1} W_{hh}^\top (a^{-1}(v_t) - W_{xh}x_t - b_h).$$

If $v_t \notin a(\mathbb{R}^p)$, the minimizer of (4) is obtained by first projecting $v_t$ onto $a(\mathbb{R}^p)$, before inverting the linear operation. To account for non-invertible matrices $W_{hh}$, we also add a regularization in the computation of the inverse. Overall we consider approximating the inverse of the layer by a regularized inverse of the form

$$f_{\theta,t}^{-1}(v_t) = (W_{hh}^\top W_{hh} + r\,\mathrm{I})^{-1} W_{hh}^\top (a^{-1}(\pi(v_t)) - W_{xh}x_t - b_h),$$

with $r > 0$ and $\pi$ a projection onto $a(\mathbb{R}^p)$.

**Regularized inversion vs. parameterized inversion.** Bengio (2014); Manchev & Spratling (2020) parameterize the inverse as a reverse layer such that

$$f_{\theta,t}^{-1}(v_t) = \psi_{\theta',t}(v_t) := a(W_{xh}x_t + Vv_t + c),$$

and learn the parameters $\theta' = (V, c)$ for this reverse layer to approximate the inverse of the forward computations. The parameterized layer needs to be learned to get a good approximation which involves numerically solving an optimization problem for each layer. These optimization problems come with a computational cost that can be better controlled by using regularized inversions presented earlier.

However, the approach based on parameterized inverses may lack theoretical grounding, as pointed out by Bengio (2020), as we do not know how close the learned inverse is to the actual inverse throughout the training process. In contrast, the regularized inversion (4) is less *ad hoc* and clearly defined and, as we shall show in the experiments, leads to competitive performance on real datasets.

In any case, the analytic formulation of the inverse gives simple insights on an approach with parameterized inverses. Namely, the analytical formula suggests parameterizing the *reverse layer* s.t. (i) the reverse activation is defined as the inverse of the activation and not any activation, (ii) the layer uses a non-linear operation followed by a linear one instead of the usual scheme, i.e., a linear operation followed by a non-linear one.

## 2.2 Linearized Inversion

Earlier instances of target propagation used direct inverses of the network layers such that the target propagation update formula would read $v_{t-1} = f_{\theta,t}^{-1}(v_t)$ in (2). Yet, we are unaware of a successful implementation of TP using directly the inverses. To circumvent this issue, Lee et al. (2015) proposed the *difference target propagation* formula that back-propagates the targets as

$$v_{t-1} = h_{t-1} + f_{\theta,t}^{-1}(v_t) - f_{\theta,t}^{-1}(h_t).$$

If the inverses were exact, the difference target propagation formula would reduce to $v_{t-1} = f_{\theta,t}^{-1}(v_t)$. Lee et al. (2015) introduced the difference target propagation formula to mitigate the approximation error of the inverses by parameterized layers. In practice, difference-type target propagation appears to be the only known successful implementation of target propagation we are aware of. The difference target propagation formula can naturally be interpreted as an approximation of the linearization used in (2), as

$$f_{\theta,t}^{-1}(v_t) - f_{\theta,t}^{-1}(h_t) = \partial_h f_{\theta,t}^{-1}(h_t)^\top (v_t - h_t) + O(\|v_t - h_t\|_2^2),$$

where $\partial_h f_{\theta,t}^{-1}(h_t)^\top$ denotes the Jacobian of the inverse of the layer at $h_t$.

We show in Appendix D that the first-order approximation we propose (2) leads to slightly better training curves than the finite-difference approximation. Moreover, our interpretation illuminates the "mystery" of this formula, which appeared to be critical to the success of target propagation.

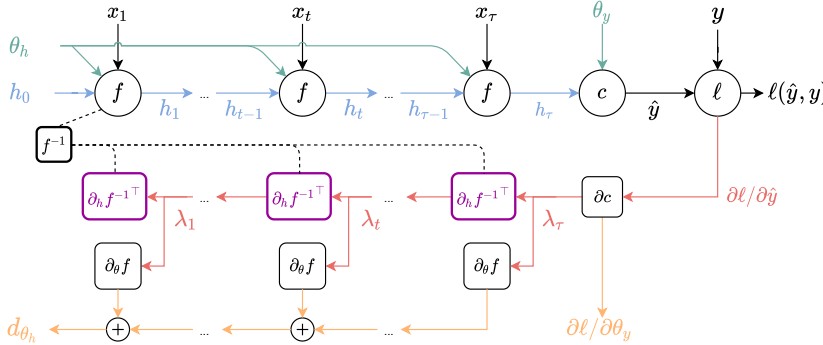

Fig. 2: The graph of computations of target propagation is the same as the one of gradient back-propagation except that $f^{-1}$ needs to be computed and Jacobian of the inverses, $\partial_h f^{-1\,\top}$ are used instead of gradients $\partial_h f$ in the transition operations.

## 3 GRADIENT BACK-PROPAGATION VERSUS TARGET PROPAGATION

**Graph of computations.** Gradient back-propagation and target propagation both compute a descent direction for the objective at hand. The difference lies in the oracles computed and stored in the forward pass, while the graph of computations remains the same. To clarify this view, we reformulate target propagation in terms of displacements $\lambda_t := v_t - h_t$ such that Eq. (1), (2) and (3) read

$$\lambda_\tau = -\gamma_h \partial_h \ell(y, c_\theta(h_\tau)), \qquad \lambda_{t-1} = \partial_h f_{\theta,t}^{-1}(h_t)^\top \lambda_t, \quad \text{for } t \in \{\tau, \dots, 1\},$$

$$d_{\theta_h} = \sum_{t=1}^\tau \partial_{\theta_h} f_{\theta,t}(h_{t-1})\lambda_t, \qquad \theta_h^{\text{next}} = \theta_h + \gamma_h d_{\theta_h}.$$

Target propagation amounts then to computing a descent direction $d_{\theta_h}$ for the parameters $\theta_h$ with a graph of computations, illustrated in Fig. 2, analogous to that of gradient-back-propagation illustrated in Appendix A. The difference lies in the use of the Jacobian of the inverse

$$\partial_h f_{\theta,t}^{-1}(h_t)^\top \quad \text{instead of} \quad \partial_h f_{\theta,t}(h_{t-1}).$$

The implementation of TP with the formula (2) can be done in a differentiable programming framework, where, rather than computing the gradient of the layer, one evaluates the inverse and keep the Jacobian of the inverse. With the precise graph of computation of TP and BP, we can compare their computational complexity explicitly and bound the difference of the directions they output.

**Arithmetic complexity.** Clearly, the space complexities of gradient back-propagation (BP) and our implementation of target propagation (TP) are the same since the Jacobians of the inverse, and the original gradients have the same size. In terms of time complexity, TP appears at first glance to introduce an important overhead since it requires the computation of some inverses. However, a close inspection of the formula of the regularized inverse reveals that a matrix inversion needs to be computed only once for all time steps. Therefore the cost of the inversion may be amortized if the length of the sequence is particularly long.

Formally, the time complexity of the forward-backward pass of gradient back-propagation is essentially driven by matrix-vector products, i.e.,

$$\mathcal{T}_{\text{BP}} = \sum_{t=1}^\tau \left[ \underbrace{\mathcal{T}(f_{\theta,t}) + \mathcal{T}(\partial_h f_{\theta,t}) + \mathcal{T}(\partial_{\theta_h} f_{\theta,t})}_{\text{Forward}} + \underbrace{\mathcal{T}(\partial_h f_{\theta,t}(h_{t-1})) + \mathcal{T}(\partial_\theta f_{\theta,t}(h_{t-1}))}_{\text{Backward}} \right]$$

$$\approx \tau(dp + p^2 + pq) + \tau(p^2 + pq),$$

where $d$ is the dimension of the input $x_t$, $q$ is the dimension of the parameters $\theta_h$, for a function $f$ we denote by $\mathcal{T}(f)$ the time complexity to evaluate $f$ and we consider e.g. $\partial_\theta f_{\theta,t}(h_{t-1}))$ as the linear function $\lambda \to \partial_\theta f_{\theta,t}(h_{t-1})\lambda$.

On the other hand, the time complexity of target propagation is

$$\mathcal{T}_{\text{TP}} = \sum_{t=1}^{\tau} \left[ \underbrace{\mathcal{T}(f_{\theta,t}) + \mathcal{T}(f_{\theta,t}^{-1}) + \mathcal{T}(\partial_{\theta_h} f_{\theta,t}) + \mathcal{T}(\partial_h f_{\theta,t}^{-1})}_{\text{Forward}} + \underbrace{\mathcal{T}(\partial_h f_{\theta,t}^{-1})(h_t)^\top) + \mathcal{T}(\partial_\theta f_{\theta,t}(h_{t-1}))}_{\text{Backward}} \right]$$
$$+ \mathcal{P}(f_{\theta,t}^{-1}),$$

where $\mathcal{P}(f_{\theta,t}^{-1})$ is the cost of encoding the inverse, which, in our case, amounts to the cost of encoding $g_\theta : z \to (W_{hh}^\top W_{hh} + r\,\mathrm{I})^{-1} W_{hh}^\top$, such that our regularized inverse can be computed as $f_{\theta,t}^{-1}(v_t) = g_\theta(a^{-1}(v_t) - W_{xh} x_t + b_h)$. Encoding $g$ comes at the cost of inverting one matrix of size $p$. Therefore, the time-complexity of target propagation can be estimated as

$$\mathcal{T}_{\text{TP}} \approx p^3 + \tau(dp + p^2 + pq) + \tau(p^2 + pq) \approx \mathcal{T}_{\text{BP}} \quad \text{if } \tau \geq p,$$

i.e., for long sequences whose length is larger than the dimension of the hidden states, the cost of TP with regularized inverses is approximately the same as the cost of BP. If a parameterized inverse was used rather than a regularized inverse, the cost of encoding the inverse would correspond to the cost of updating the reverse layers by, e.g., a stochastic gradient descent. This update has a cost similar to BP. However, it is unclear whether these updates get us close to the actual inverses.

**Bounding the difference between target propagation and gradient back-propagation.** As the computational graphs of BP and TP are the same, we can bound the difference between the oracles returned by both methods. First, note that the updates of the parameters of the output functions are the same since, in TP, gradients steps of the loss are used to update these parameters. The difference between TP and BP lies in the updates with respect to the parameters of the transition operations. For BP, the updates are computed by chain rule as

$$\partial_{\theta_h} \ell(y, g_\theta(x_{1:\tau})) = \sum_{t=1}^{\tau} \partial_{\theta_h} f_{\theta,t}(h_{t-1}) \frac{\partial h_\tau}{\partial h_t} \partial_h \ell(y, c_\theta(h_\tau)),$$

where the term $\partial h_\tau / \partial h_t$ decomposes along the time steps as $\partial h_\tau / \partial h_t = \prod_{s=t+1}^{\tau} \partial_h f_{\theta,s}(h_{s-1})$. The direction computed by TP has the same structure, namely it can be decomposed for $\gamma_h = 1$ as

$$d_\theta = \sum_{t=1}^{\tau} \partial_{\theta_h} f_{\theta,t}(h_{t-1}) \frac{\hat{\partial} h_\tau}{\hat{\partial} h_t} \partial_h \ell(y, c_\theta(h_\tau)),$$

where $\hat{\partial} h_\tau / \hat{\partial} h_t = \prod_{s=t+1}^{\tau} \partial_h f_{\theta,s}^{-1}(h_s)^\top$. We can then bound the difference between the directions given by BP or TP as, for any matrix norm $\| \cdot \|$ as formally stated in the following lemma.

**Lemma 3.1.** *The difference between the oracle returned by gradient back-propagation* $\partial_{\theta_h} \ell(y, g_\theta(x_{1:\tau}))$ *and the oracle returned by target propagation can be bounded as*

$$\| \partial_{\theta_h} \ell(y, g_\theta(x_{1:\tau})) - d_\theta \| \leq c \sup_{t=1,\dots,\tau} \| \partial_h f_{\theta,t}(h_{t-1}) - \partial_h f_{\theta,t}^{-1}(h_t)^\top \|,$$

*where* $c = \sum_{t=1}^{\tau} \sum_{s=0}^{t-1} a^s b^{t-1-s}$ *with* $a = \sup_{t=1,\dots\tau} \| \partial_h f_{\theta,t}(h_{t-1}) \|, b = \sup_{t=1,\dots\tau} \| \partial_h f_{\theta,t}^{-1}(h_t)^\top \|.$

*For regularized inverses, we have, denoting* $u_t = W_{xh} x_t + W_{hh} h_{t-1} + b_h,$

$$\| \partial_h f_{\theta,t}(h_{t-1}) - \partial_h f_{\theta,t}^{-1}(h_t)^\top \| \leq \| W_{hh}^\top \| \left( \| \nabla a(u_t) - \nabla a(u_t)^{-1} \| + \| \mathrm{I} - (W_{hh}^\top W_{hh} + r\,\mathrm{I})^{-1} \| \| \nabla a(u_t)^{-1} \| \right).$$

For the two oracles to be close, we then need the preactivation $u_t = W_{xh} x_t + W_{hh} h_{t-1} + b_h$ to lie in the region of the activation function that is close to being linear s.t. $\nabla a(u_t) \approx \mathrm{I}$. We also need $(W_{hh}^\top W_{hh} + r\,\mathrm{I})^{-1}$ to be close to the identity which can be the case if, e.g., $r = 0$ and the weight matrices $W_h$ were orthonormal. By initializing the weight matrices as orthonormal matrices, the differences between the two oracles can be closer. However, in the long term, target propagation appears to give better oracles, as shown in the experiments below.

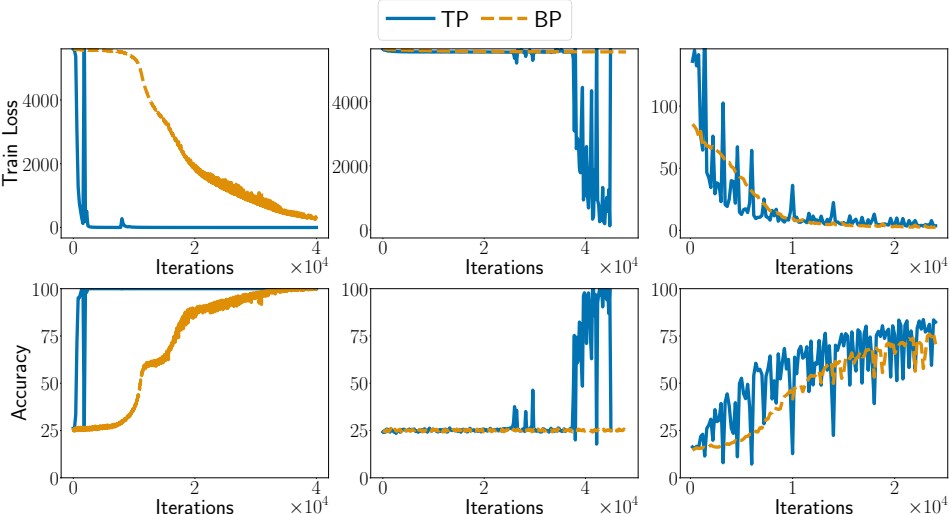

Fig. 3: Temporal order problem $T = 60$, Temporal Problem $T = 120$, Adding problem $T = 30$.

**Target propagation as a Gauss-Newton method?**  Recently target propagation has been interpreted as an approximate Gauss-Newton method, by considering that the difference target propagation formula approximates the linearization of the inverse, which itself is a priori equal to the inverse of the gradients (Bengio, 2020; Meulemans et al., 2020; 2021). Namely, provided that $f_{\theta,t}^{-1}(f_{\theta,t}(h_{t-1})) \approx h_{t-1}$ such that $\partial_h f_{\theta,t}(h_{t-1}) \partial_h f_{\theta,t}^{-1}(h_t) \approx \mathrm{I}$, we have

$$\partial_h f_{\theta,t}^{-1}(h_t) \approx (\partial_h f_{\theta,t}(h_{t-1}))^{-1}.$$

By composing the inverses of the gradients, we get an update similar to the one of Gauss-Newton (GN) method. Namely, recall that if $n$ invertible functions $f_1, \ldots, f_n$ were composed to solve a least square problem of the form $\|f_n \circ \ldots \circ f_1(x) - y\|_2^2$, a Gauss-Newton update would take the form $x^{(k+1)} = x^{(k)} - \partial_{x_0} f_1(x_0)^{-\top} \partial_{x_1} f_2(x_1)^{-\top} \ldots \partial_{x_{n-1}} f(x_{n-1})^{-\top} (x_n - y)$ where $x_t$ is defined iteratively as $x_0 = x^{(k)}$, $x_{t+1} = f_t(x_t)$. In other words, GN and TP share the idea of composing the inverse of gradients. However, numerous differences remain: (i) in e.g. RNNs, the gradients w.r.t. to the weights are defined as a sum of the composition of the gradients and the inverse of this sum is a priori not the sum of the inverses, (ii) if the real rationale of GN was used, the gradients w.r.t. the loss should also be inverted, and the gradients w.r.t. to the weights should also be inverted which is not the case in the usual implementation of TP Lee et al. (2015); Bengio (2020). Even if TP was approximating GN, it is unclear whether GN updates are adapted to stochastic problems.

## 4 EXPERIMENTS

In the following, we compare our simple target propagation approach, which we shall refer to as **TP**, to gradient Back-Propagation referred to as **BP**. We follow the experimental benchmark of Manchev & Spratling (2020) to which we add results on RNNs on CIFAR and GRUs on FashionMNIST. Additional experiments, details on the initialization and the hyper-parameter selection can be found in Appendix D.

**Data.**  We consider two synthetic datasets generated to present training difficulties for RNNs and several real datasets consisting of scanning images pixel by pixel to classify them (Hochreiter & Schmidhuber, 1997; Le et al., 2015; Manchev & Spratling, 2020).

*Temporal order problem.*  A sequence of length $T$ is generated using a set of randomly chosen symbols $\{a, b, c, d\}$. Two additional symbols $X$ and $Y$ are added at positions $t_1 \in [T/10, 2T/10]$ and $t_2 \in [4T/10, 5T/10]$. The network must predict the correct order of appearance of $X$ and $Y$ out of four possible choices $\{XX, XY, YX, YY\}$.

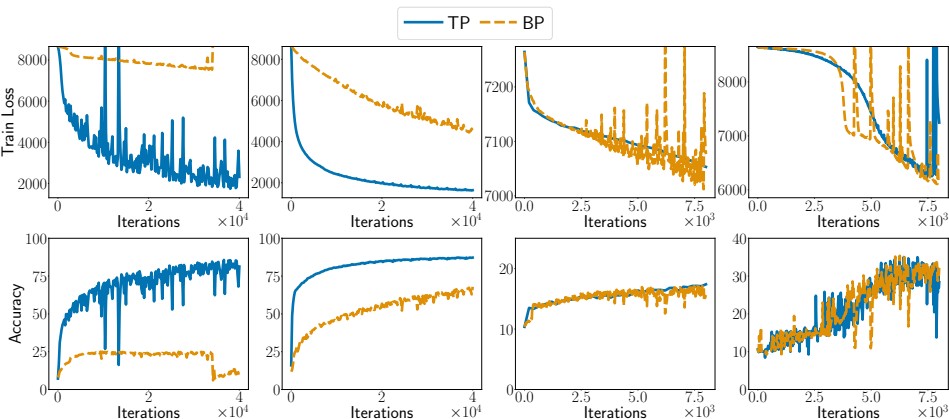

Fig. 4: Image classification pixel by pixel. From left to right: MNIST, MNIST with permuted images, CI-FAR10, FashionMNIST with GRU.

*Adding problem.* The input consists of two sequences: one is made of randomly chosen numbers from $[0, 1]$, and the other one is a binary sequence full of zeros except at positions $t_1 \in [1, T/10]$ and $t_2 \in [T/10, T/2]$. The second position acts as a marker for the time steps $t_1$ and $t_2$. The goal of the network is to output the mean of the two random numbers of the first sequence $(X_{t_1} + X_{t_2})/2$.

*Image classification pixel by pixel.* The inputs are images of (i) grayscale handwritten digits given in the database MNIST (LeCun & Cortes, 1998), (ii) colored objects from the database CIFAR10 (Krizhevsky, 2009) or (iii) grayscale images of clothes from the database FashionM-NIST (Xiao et al., 2017). The images are scanned pixel by pixel and channel by channel for CI-FAR10, and fed to a sequential network such as a simple RNN or a GRU network (Cho et al., 2014). The inputs are then sequences of $28 \times 28 = 784$ pixels for MNIST or FashionMNIST and $32 \times 32 \times 3 = 3072$ pixels for CIFAR with a very long-range dependency problem. We also consider permuting the images of MNIST by a fixed permutation before feeding them into the network, which gives potentially longer dependencies in the sequential data.

**Model.** In both synthetic settings, we consider randomly generated mini-batches of size 20, a simple RNN with hidden states of dimension 100, and hyperbolic tangent activation. For the temporal order problem, the last layer uses a soft-max function on top of a linear operation, and the loss is the cross-entropy. For the adding problem, the last layer is linear, the loss is the mean-squared error, and a sample is considered to be accurately predicted if the mean squared error is less than 0.04 as done by (Manchev & Spratling, 2020).

For the classification of images with sequential networks, we consider mini-batches of size 16 and a cross-entropy loss. For MNIST and CIFAR, we consider a simple RNN with hidden states of dimension 100, hyperbolic tangent activation, and a softmax output. For FashionMNIST, we consider a GRU network and adapted our implementation of target propagation to that case while using hidden states of dimension 100 and a softmax output.

**Target propagation can tackle long sequences better than gradient back-propagation.** In Fig. 3, we observe that TP performs better than BP on the temporal ordering problem: it is able to reach 100% accuracy in fewer iterations than BP for sequences of length 60 and, for sequences of length 120, it is still able to reach 100% accuracy in fewer than 40 000 iterations while BP is not. On the other hand, for the adding problem, TP performs less well than BP. The contrast in performance between the two synthetic tasks was also observed by (Manchev & Spratling, 2020) using difference target propagation with parameterized inverses. The main difference between these tasks is the different nature of the outputs, which are binary for the temporal problem and continuous for the adding problem.

In Fig. 4, we observe that TP generally performs better than BP for image classification tasks. For the MNIST dataset, it reaches around 74% accuracy after $4 \cdot 10^4$ iterations. This phenomenon is also

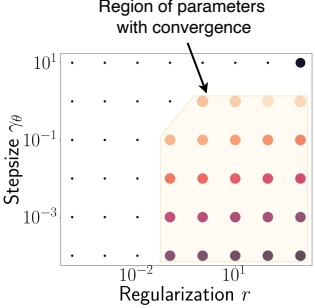
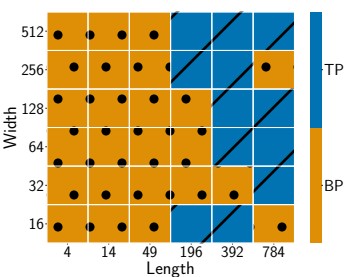

(6a) Conv. w.r.t. stepsize & regularization      (6b) Perf. vs width & length.

observed with permuted images, where the optimization appears smoother, and TP obtains around 86% accuracy after $4 \cdot 10^4$ iterations and is still faster than BP. On the CIFAR dataset, no algorithms appear to reach a significant accuracy, though TP is still faster. On the FashionMNIST dataset, where a GRU network is used, our implementation of TP performs on par with BP, which shows that our approach can be generalized to more complex networks than a simple RNN.

**Target propagation requires a non-zero regularization term.** As mentioned in Sec. 3, by using an analytical formula to compute the inverse of the layers, we can question the interpretation of TP as a Gauss-Newton method, which would amount to TP without regularization. To understand the effect of the regularization term, we computed the area under the training loss curve of TP for 400 iterations on a $\log_{10}$ grid of varying step-sizes $\gamma_\theta$ and regularizations $r$ for a fixed $\gamma_h = 10^{-3}$. The results are presented in Fig. 6a, where the smaller the area, the brighter the point and the absence of dots in the grid mean that the algorithm diverged. Fig. 6a shows that without regularization we were not able to obtain convergence of the algorithm. Simply using the gradients of the inverse as in a Gauss-Newton method may not directly work for RNNs. Additional modifications of the method could be added to make target propagation closer to Gauss-Newton, such as inverting the layers with respect to their parameters as proposed by Bengio (2020). For now, the regularization appears to successfully handle the rationale of target propagation.

**Target propagation is adapted for long sequences.** In Fig. 6b, we compare the performance of BP and TP in terms of accuracy after 400 iterations on the MNIST problem for various widths determined by the size of the hidden states and various lengths determined by the size of the inputs (i.e., we feed the RNN with $k$ pixels at a time, which gives a length $784/k$). Fig 6b shows that TP is generally appropriate for long sequences, while BP remains more efficient for short sequences. TP can then be seen as an interesting alternative for dynamical problems which involve many discretization steps as in RNNs and related architectures.

## CONCLUSION

We proposed a simple target propagation approach grounded in two important computational components, regularized inversion, and linearized propagation. The proposed approach also sheds light on previous insights and successful rules for target propagation. We will publicly release our code to facilitate the reproduction of the results. We have used target propagation within a stochastic gradient outer loop to train neural networks for a fair comparison to stochastic gradient using gradient backpropagation. Developing adaptive stochastic gradient algorithms in the spirit of Adam that lead to boosts in performance when using target propagation instead of gradient backpropagation is an interesting avenue for future work. Continuous counterparts of target propagation in a neural ODE spirit is also an interesting avenue for future work.

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
