# OpenReview forum: "Target Propagation via Regularized Inversion"
_ICLR.cc/2022/Conference — ICLR 2022 Submitted_

### Official Review · Reviewer_239T · 2021-11-01

**Correctness:** 3
**Technical Novelty And Significance:** 3
**Empirical Novelty And Significance:** 2
**Recommendation:** 5
**Confidence:** 4

**Main Review:**

The primary strengths of this manuscript are that it is easy to read, the proposed algorithm is straightforward to implement in the current computational setup, and the improvements versus gradient descent.

In my opinion, there are two main weaknesses of the manuscript.

(1) The theory needs to be more fully explored and discussed.  For example, Lemma 3.1 bounds the different between the direction computed by TP and BP. First, it should be noted that this bound is quite loose in the case where the constants $a$ or $b$ are greater than 1. Second, there needs to be more complete discussion of this meaning of this Lemma, as the argument that TP is better than BP depends on the fact that TP calculates a \textit{different} direction than BP.  I would appreciate greater follow-up and discussion in the analysis on how TP and BP differ rather than on how similar they are. Next, I find the relationship to Gauss-Newton methods quite interesting, but that section is barely sketched out and does not seem fully derived.  How close is TP to a Gauss-Newton method?  Also, I found the phrase "In our experiments, we have not been able to get a non-diverging sequence of iterates if the regularization was set to 0, which questions the interpretation of target propagation as a Gauss-Newton method" peculiar.  In a Gauss-Newton method, you need more samples than parameters to get the Jacobian to be well-conditioned and calculate a good direction, or it has to be regularized somehow.  The fact that you have to regularize the TP technique seems consistent with a Gauss-Newton method.  Am I missing something?  I believe that fully-fleshing out this relationship, if indeed they are close, would help provide rationale for your method.

(2) The empirical evidence, in my opinion, is lacking.  There are few experiments and runs here.  When does TP outperform GD?  Why does it outperform GP?  This manuscript needs to explain more about what is different in the optimization to make it work better, and evaluate how  robust those improvements are with much more detailed and extensive experimentation, especially since the improvements are not fully explained from the theoretical side.  Additionally, there are many, many, learning algorithms designed to improve learning in RNNs.  There needs to be more comparison and discussion of these algorithms.  For example, how does this compare to the algorithm proposed in Lee et al 2015?  How does this relate to Hessian Free-Optimization (e.g., [1]), which is somewhat alluded to with the relationship to Gauss-Newton?  There must be a more complete literature search and discussion on optimizing RNNs as a whole, rather than the relatively narrow focus on exclusively TP.

[1] Martens, James, and Ilya Sutskever. "Learning recurrent neural networks with hessian-free optimization." ICML. 2011.

**Summary Of The Paper:**

This manuscript proposes a new way of approaching optimization in RNNs based on Target Propagation.  It develops a linearized approximation to the method to make it straightforward to implement within a standard computational graph, and develops some theory about the approximations, and shows improvement compared to gradient descent in a few experiments.

**Summary Of The Review:**

This manuscript proposes an interesting and potentially useful algorithm, but needs to more fully explain its theoretical properties and provide more detailed experimentation and comparisons.

---

### Official Review · Reviewer_Dczs · 2021-11-01

**Correctness:** 3
**Technical Novelty And Significance:** 2
**Empirical Novelty And Significance:** 2
**Recommendation:** 5
**Confidence:** 3

**Main Review:**

The paper presents their model quite clear. Demonstration and comparison with traditional BP algorithm is illustrative. The target propagation algorithm is an interesting new directions that worth exploring.

The main downside of the paper would be experimental demonstration.

-- It's good to have experiments on realistic image dataset such as cifar10, however, either the baseline as well as proposed model seems to be quite weak (their accuracy is quite low, fig 4, less than 20%). I understand the authors focused on the illustrative purpose, but that also makes it unclear how the model could generalize,  scale and apply to the realistic datasets. Since the current literature has quite a few competitive RNN models on image classification, why not directly take their models and replace the BP component to see how it works? I'm not asking for the SOTA performance, only the competitive results could be good enough to show the potential of the proposed work.

-- I assume the framework is targeted on recurrent model training, perhaps the authors would make it more clear throughout the paper. Some of the places claim such as " We have used target propagation within a stochastic gradient outer loop to train neural networks for a fair compar- ison to stochastic gradient using gradient backpropagation." (in the conclusion section), which might be misleading or over claimed. If in general the proposed model is flexible enough to handle all kinds of neural networks, perhaps experiments should also include convolution neural net etc for the illustration.

**Summary Of The Paper:**

The paper proposes the train the target propagation algorithm (as an alternative to gradient back-propagation algorithm) through regularized inversion. Specifically, they discussed two techniques including regularized inversion and linearize propagation, and demonstrate their effectiveness (via convergence speed and accuracy) for learning the RNN model on synthetic and image benchmarks.

**Summary Of The Review:**

The paper explore an interesting alternative to BP. However, the current form didn't fully convince me its superiority over bp. I lean towards the rejection, but willing to adjust my rating based on the authors' feedbacks.

---

### Official Review · Reviewer_Q1ay · 2021-11-02

**Correctness:** 4
**Technical Novelty And Significance:** 2
**Empirical Novelty And Significance:** 3
**Recommendation:** 6
**Confidence:** 3

**Main Review:**

**Strengths:**
- The proposed scheme features approximating the inverse of the layer by a regularized inverse that facilitates the use of autograd.
- Linearized inversion offers a natural interpretation as an approximation of difference target propagation.
- The cost of matrix inversion can be amortized by the length of sequence.
- There is theoretical guarantee available that bounds the difference between target propagation and backpropagation.

**Weaknesses:**
- The process seems to have large variation on training loss and accuracy compared to back propagation, which might be an indication that the scheme is less amendable to theoretical analysis.

**Correctness:**
- There is no false claims to the best of my knowledge

**Clarity:**
- The paper is clear and structured
- A typo in the paragraph under equation (2): "allow us tp interpret" ---> "allow us to interpret"

**Additional Comments:**
- Is there any convergence result possible for the proposed target propagation scheme given that the backward updates are more tractable compared to the original version? For example, there is a convergence analysis for feedback alignment by Song et al.. Anything similar possible for target propagation?

**Summary Of The Paper:**

The authors provide a modified version of target propagation that is easily implementable via autograd system. This paper also compares the computation complexity of the algorithm with backpropagation and shows empirically that the algorithm yields good performance on recurrent neural networks with long sequences.

**Summary Of The Review:**

The target propagation scheme proposed in this paper look novel to me. Even though the numerical result is a bit confusing, the algorithm has natural interpretation and more computationally tractable.

---

### Official Review · Reviewer_NQCB · 2021-11-02

**Correctness:** 3
**Technical Novelty And Significance:** 2
**Empirical Novelty And Significance:** 2
**Recommendation:** 3
**Confidence:** 3

**Main Review:**

While I find target propagation (TP) interesting, I am not very enthusiastic about the current paper. As detailed below, the main aspects that attract me towards TP are absent from the proposed algorithm, and its merits as a generic drop-in replacement for backpropagation are not strong enough or at least not yet well argued for.

The authors propose a new prescription to compute targets in TP. Unlike much of the recent work on TP, this prescription leads to a parameter update that requires weight transport. This limits its interest and narrows the scope of the contributions, as optimization without weight transport is of great interest not just in theoretical neuroscience but also for neuromorphic hardware design and large-scale distributed network implementations.

As noted by the authors, the proposed algorithm is also strictly costlier than backpropagation, although the relative difference in cost becomes smaller as sequence length grows. It also does not allow for decoupled updates, or at least this is not commented in the paper. Its applicability to decentralized optimization in distributed settings (another setting of potential interest for TP algorithms) is therfore not clear.

For the proposed algorithm to be interesting, it therefore has to produce descent directions that lead to faster optimization than gradients. The paper does not provide any theoretical guarantees that this is the case. The merits of the new algorithm must then be judged by the experimental results. This is a very difficult question to answer experimentally, and I did not find the results extremely convincing, but I am admittedly not an expert in sequence learning problems.

I have a number of questions for the authors:
- In page 3, below eq. 2, it is mentioned that DTP is a 0-th order approximation. Can the authors clarify this remark? In my understanding, the difference correction of DTP stems from a first-order expansion of $f$, see Appendix B of Lee et al. (2015). In what sense is it a 0-th order correction?

- What are the new insights provided in section 2.2 compared to what is already used by Lee et al. to arrive at DTP (and perhaps more clearly restated in Lemma 1 of Meulemans et al. 2020)?

-  I'm having trouble with eq. 4 and the text that follows. What happens when there are multiple minimizers of the objective presented in (4), for example when $a$ is the ReLU activation? How should the next unnumbered equation appearing below "Formally, for [...] the ReLU, their inverse can be obtained analytically" be understood in this case?

- There are numerous remarks on the vanishing/exploding gradient problem and allusions to the promise of solving this problem with TP. But since the proposed parameter update relies on products of Jacobian inverses, isn't it also subject to vanishing and exploding gradient problems?

- Why is the relationship between TP and Gauss-Newton optimization pointed out by Bengio (2020) and Meulemans et al. (2020) called into question due to the difficulties observed when setting the regularizer to zero?

- Is GEMINI (LeCun et al., 1989), cited already in the abstract, the desired reference? To me, GEMINI is more obviously related to synthetic gradients than target propagation, and it is not usually cited as LeCun's original TP paper. Reference [1] below is perhaps more appropriate.

[1] LeCun (1986) "Learning processes in an asymmetric threshold network"

**Summary Of The Paper:**

The authors study a variant of target propagation in which targets are computed by solving a sequence of minimization problems. Instead of resorting to iterative methods the authors propose to use an analytical solution. The algorithm is investigated as a recurrent neural network learning algorithm in a number of experiments. A relationship between the proposed descent direction and the loss function gradient is developed.

**Summary Of The Review:**

The algorithm loses most of the features that make target propagation attractive as an alternative to backpropagation. As an optimization algorithm, its guarantees are not sufficiently strong.

---

### Decision · Program_Chairs · 2022-01-20

**Decision:**

Reject

**Comment:**

The paper proposes a new approach to target propagation that performs well when used in RNNs on sequence modelling. The paper falls into something of an uncanny valley, where it is different enough from the original TP to lose some of its motivation ("biological plausibility"), and is now directly competing with backprop. Claims about outperforming backprop require EXTREMELY thorough and rigorous experimental evidence. Without meaning to cast any doubt on the authors work, there have simply been a lot of papers over the years that saw some improvements over backprop in some setting, that have not generalised or even been reproducible.